# The Impact of Historical Land Use Change From 1850 to 2000 on Secondary Particulate Matter and Ozone

Colette L. Heald[1], Jeffrey A. Geddes[2]

[1]Department of Civil and Environmental Engineering, Massachusetts Institute of Technology, Cambridge, MA, USA
[2]Department of Earth and Environment, Boston University, Boston, MA, USA

*Correspondence to*: Colette L. Heald (heald@mit.edu)

**Abstract.** Anthropogenic land use change (LUC) since pre-industrial (1850) has altered the vegetation distribution and density around the world. We use a global model (GEOS-Chem) to assess the attendant changes in surface air quality and the direct radiative forcing (DRF). We focus our analysis on secondary particulate matter and tropospheric ozone formation. The
general trend of expansion of managed ecosystems (croplands and pasturelands) at the expense of natural ecosystems has led to an 11% decline in global mean biogenic volatile organic compound emissions. Concomitant growth in agricultural activity has more than doubled ammonia emissions and increased emissions of nitrogen oxides from soils by more than 50%. Conversion to croplands has also led to a widespread increase in ozone dry deposition velocity. Together these changes in biosphere-atmosphere exchange have led to a 14% global mean increase in biogenic secondary organic aerosol (BSOA)
surface concentrations, a doubling of surface aerosol nitrate concentrations, and local changes in surface ozone of up to 8.5 ppb. We assess a global mean LUC-DRF of +0.017 $Wm^{-2}$, -0.071 $Wm^{-2}$, and -0.01 $Wm^{-2}$ for BSOA, nitrate, and tropospheric ozone, respectively. We conclude that the DRF and the perturbations in surface air quality associated with LUC (and the associated changes in agricultural emissions) are substantial and should be considered alongside changes in anthropogenic emissions and climate feedbacks in chemistry-climate studies.

**1 Introduction**

Humans have dramatically altered the land surface of the Earth, affecting over half of the land surface and permanently clearing over ¼ of the planet's forests (Hurtt et al., 2006; Vitousek et al., 1997). Land use changes have accelerated with population growth, with 64% of cropland growth occurring since 1850 (Hurtt et al., 2011). These substantial shifts in land use have perturbed the exchange of carbon, water, and energy between the biosphere and atmosphere, impacting weather,
and climate (Pielke et al., 2002; Pitman et al., 2009). Land use change also alters the biosphere-atmosphere exchange of gases and particles that impact air quality and contribute to short-lived radiative forcing, however few studies have quantified these effects (Heald and Spracklen, 2015).

Particulate matter (PM) and tropospheric ozone are deleterious to human health and dominate uncertainty in current estimates of global climate forcing (IPCC, 2013). Air pollution is the leading environmental cause of premature mortality

world-wide (OECD, 2012); exposure to ambient PM and surface ozone was responsible for over 3.7 million premature deaths in 2010 (Lim et al., 2013). The most recent IPCC estimates that tropospheric ozone and aerosols contribute +0.40 $Wm^{-2}$ and -0.35 $Wm^{-2}$ respectively to global direct radiative forcing (DRF) (IPCC, 2013). PM and ozone are short-lived climate pollutants, with lifetimes of about a week and about a month, respectively (Balkanski et al., 1993; Young et al.,
2013). As a result, reductions in the concentrations of the warming components (ozone, black carbon) may be an effective strategy for mitigating near-term climate change (Shindell et al., 2012). At the same time, the short lifetimes of these species coupled with the multitude of physical and chemical sources, limits confidence in estimated global climate forcing from these species (Myhre et al., 2013; Stevenson et al., 2013). In addition, the impact of anthropogenic land use change are not included in these estimates of PM and tropospheric ozone radiative forcing.

The terrestrial biosphere is a source of organics and nitrogen oxides (NOx) that can contribute to PM and ozone formation. Biogenic volatile organic compounds (BVOC) emitted from vegetation, such as isoprene and monoterpenes, react quickly in the atmosphere to form low-volatility vapours that can condense to the particle phase and produce secondary organic aerosol (SOA) (Hallquist et al., 2009). Given sufficient NOx, this oxidation of BVOCs can also produce ozone. However, in clean, NOx-poor conditions, these BVOCs can react with and therefore consume ozone (Wang and Shallcross, 2000). The emission
of BVOC depends strongly on the type and density of vegetation (Guenther et al., 2012). Similarly, microbial sources of nitrogen oxides from soils vary with land use, and with canopy density (Hudman et al., 2012). Managed ecosystems, such as croplands and pasturelands, are the dominant source of ammonia emissions to the atmosphere through emissions from fertilizer and domesticated animals (Erisman et al., 2008). In combination with nitric acid formed from the oxidation of NOx, ammonia can produce ammonium nitrate, an increasingly important source of inorganic PM in regions where sulfur
emissions controls substantially reduce sulfate (Paulot et al., 2016; Pinder et al., 2007). The terrestrial biosphere is also a sink of gases and particles. In particular, the dry deposition of ozone at the surface accounts for ~20% of the ozone loss in the troposphere (Stevenson et al., 2006). This removal is most efficient over high density vegetation and croplands via stomatal uptake. Perturbation to vegetation and transitions between land types alter these fluxes, with implications for PM and ozone.

Tree mortality, for example associated with insect infestation or disease, can modulate biosphere-atmosphere exchange, generating transitory perturbations in air quality (Berg et al., 2013; Geddes et al., 2016). However, conversion of land cover, for example via clearing, can lead to long-term changes in surface properties and therefore atmospheric composition. A number of studies have explored how both natural and anthropogenic future land use change may impact atmospheric chemistry (Ganzeveld et al., 2010; Heald et al., 2008; Wu et al., 2012). Few studies have explored the impact of historical
land use change on PM and tropospheric ozone. As a result, anthropogenic land use change is absent from most estimates of radiative forcing from aerosols and tropospheric ozone. Ward et al. (2014) investigate the impact of land use and land cover change (LULCC) on greenhouse gases (including tropospheric ozone) and aerosols. They estimate that historical changes in LULCC result in a radiative forcing of +0.12 $Wm^{-2}$ for ozone and -0.04 $Wm^{-2}$ for aerosols (-0.02 $Wm^{-2}$ direct, -0.02 $Wm^{-2}$

indirect) relative to 1850. In this study the increase in ozone associated with LULCC is largely associated with the increase in methane and fires with partial compensation due to a 6% increase in ozone dry deposition. Their estimate of aerosol forcing is not disaggregated by species, but includes dust, biogenic SOA, and smoke. Unger (2014) suggests that land use change is responsible for -0.13 Wm$^{-2}$ of radiative forcing from tropospheric ozone, and +0.09 Wm$^{-2}$ from biogenic SOA (direct only), primarily due to decreases in BVOC emissions since 1850These two assessments of radiative forcing of ozone and PM associated with land use do not agree on the sign of the forcing. However, it is critical to note that these studies differ fundamentally in design and in the processes and species considered, highlighting the complexity of this forcing and the need to quantify and compare specific impacts. A review of the potential impacts of land use change on air quality and climate, suggests that historical LULCC has led to an aerosol direct radiative cooling of ~ -0.10 Wm$^{-2}$, roughly 30% of current estimates of aerosol DRF (Heald and Spracklen, 2015). However, this review also points out the large uncertainty associated with these changes and the need for additional modelling studies on this topic. With this study, we aim to complement previous investigations and explore the impacts of historical global anthropogenic land use change on biosphere-atmosphere exchange processes and the resulting perturbations to secondary PM and ozone.

## 2 Model Description

To characterize the impact of historical land use change on air quality, we use v9-02 of the global chemical transport model GEOS-Chem (www.geos-chem.org). GEOS-Chem is driven by assimilated meteorology from the Global Modeling and Assimilation Office (GMAO). Here we use GEOS-5 meteorology for the year 2010. The native resolution (0.5°x0.67° horizontal resolution with 72 vertical levels) is degraded to 2°x2.5° and 47 vertical levels for computational efficiency.

The GEOS-Chem oxidant-aerosol simulation includes $H_2SO_4$-$HNO_3$-$NH_3$ aerosol thermodynamics described by ISORROPIA II (Fountoukis and Nenes, 2007; Pye et al., 2009) coupled to a detailed $HO_x$-$NO_x$-VOC-$O_3$ chemical mechanism. The model scheme also includes primary carbonaceous aerosols (Park et al., 2003), sea salt aerosol (Alexander et al., 2005; Jaegle et al., 2010), and soil dust (Fairlie et al., 2007; Ridley et al., 2013). Secondary organic aerosol (SOA) is produced from the oxidation of biogenic hydrocarbons, aromatics, and IVOCs and represented with a volatility basis set approach (Pye et al., 2010; Pye and Seinfeld, 2010).

In this study, global anthropogenic emissions for 1850 and 2000 follow the Representative Concentration Pathway (RCP) historical emissions dataset (van Vuuren et al., 2011) as implemented by Holmes et al. (2013). These include fossil fuel, biofuel, and agricultural emissions. Fire emissions are specified using GFED3 for the year 2010 (van der Werf et al., 2010), consistent with the meteorology, and are fixed for all simulations. Methane concentrations are similarly fixed at year 2010 levels.

We use the GEOS-Chem land use module recently developed by Geddes et al. (2016) to specify consistent surface properties and to simulate surface-atmosphere exchange processes. These include the emissions of BVOC, the emission of NOx from

soils, and dry deposition of gases and particles. The land module uses 16 plant functional types (PFT), consistent with those described by the Community Land Model (CLM) (Lawrence et al., 2011). The total leaf area index (LAI) is calculated interactively based on the PFT distribution and PFT-specific seasonal LAI taken from the CLM, derived from MODIS observations. BVOC emission factors for these PFTs are scaled online by activity factors describing emission response to

light, temperature, leaf age, and $CO_2$ following MEGAN v2.1 (Guenther et al., 2012). The PFTs are mapped to the biomes used for the soil $NO_x$ emissions scheme described by Hudman et al. (2012). This parameterization includes biome-specific emissions, as well as re-emission of wet and dry deposited nitrogen and fertilizer and manure nitrogen, all modulated online by temperature, soil moisture, and rain. Finally, dry deposition is based on the resistance-in-series scheme of Wesely (1989), with aerosol-specific deposition described by Zhang et al. (2001). The surface resistance for gases includes resistances to the

ground, lower canopy, and vegetation, all of which are driven by fixed parameters for 11 land use types specified in the original Wesely (1989) parameterization. The PFTs are mapped to these 11 land use types. In addition, the aerodynamic resistance and quasi-laminar resistance calculations were altered to use biome-specific roughness heights (which will reflect specified land use), rather than values from the assimilated meteorological product.

To estimate the shortwave and longwave flux perturbations associated with tropospheric ozone and aerosols we apply the

local (gridbox) monthly mean radiative flux-to-burden relationship for each species archived from previous simulations (Heald et al., 2014) to changes in simulated burden. The simulation of Heald et al. (2014) uses a similar version of GEOS-Chem (v9.01.03) with identical meteorology and spatial resolution to the simulations explored in this study, ensuring that this offline application of radiative efficiency is a good approximation. We note that these radiative efficiencies are estimated using present-day land reflectances. The physical and optical properties assumed for aerosol species and the RRTMG

radiative transfer model are described in Heald et al. (2014).

In this study we perform a series of simulations to explore the impact of land use change (and the associated changes in agricultural emissions) on ozone and aerosols (Table 1). All simulations are performed with year 2010 meteorology, fire emissions, and methane concentrations. Land use change modulates surface albedo, energy, and water exchange (Pielke et al., 2002; Pielke et al., 2011; Pitman et al., 2009) which may feedback on atmospheric composition (Ganzeveld et al., 2010;

Ganzeveld and Lelieveld, 2004). Unger (2014) suggest that these feedbacks are small compared to the perturbation in BVOC emissions from historical land use change. By design, by fixing meteorology at year 2010, we do not quantify these impacts in this study. Rather, our simulations focus on the direct impact of changes in biosphere-atmosphere exchange. By keeping methane concentrations constant we neglect changes in oxidative capacity driven by changes in local methane sources associated with agriculture (e.g. expansion of rice paddies, growth in livestock). Methane concentrations also do not respond

to the changes in oxidative capacity associated with land-use driven changes in short-lived precursor emissions (assessed in Section 5). Given the challenges associated with identifying dust regions produced from human-driven desertification (Ginoux et al., 2012), we keep this source constant and do not characterize the land use change impacts on dust. While land use change can produce large fire events, for example, deforestation fires associated with land clearing, these fires are

typically transitory and vary considerably year-to-year (Hansen et al., 2013; van der Werf et al., 2010). Regular fire emissions associated with land use change, such as agricultural waste burning, make up less than 5% of global annual smoke emissions (van der Werf et al., 2010). Ward et al. (2014) explore the impacts of historical LULCC impacts on dust and smoke. In this study we focus on the impact of land use change on secondary aerosol and ozone formation. We also perform
a set of simulations to separately estimate the impact of increasing agricultural emissions associated with land use change. In these simulations we assume that all changes in agricultural emissions of ammonia from 1850 to 2000 in the RCP emissions inventory are associated with land use change (i.e. conversion to either croplands or pastures). In addition, for 1850 agricultural emissions, we scale down the fertilizer source of soil NOx emissions to 15.7% of year 2000 values (equivalent to the global 1850:2000 ratio for agricultural sources of ammonia in the RCP emissions). We perform simulations to isolate the
impact of anthropogenic land use change alone, agricultural emissions changes alone, and both together as described by Table 2. We perform each set of simulations under both pre-industrial (2000) and present-day (2000) anthropogenic (non-agricultural) emissions to bracket the potential range of these impacts depending on the background atmospheric conditions. We focus our results on the net impacts of land use change along with the associated changes in agricultural emissions (which we collectively refer to as LUC), unless otherwise specified.

**3 Land Use Change from 1850 to 2000**

Figure 1 shows the present-day (2000) distribution of vegetation used here grouped from 15 vegetated PFTs to 6 main vegetation categories for simplicity. The PFT distribution for present-day is the satellite phenology dataset used by CLM4 which is based on MODIS data and cropping datasets (Lawrence et al., 2011).

Figure 2 shows the change in vegetation distribution from pre-industrial (1850) to present-day (2000) used here. The
historical (1850) PFT distribution is specified as the Lawrence et al. (2012) CLM-specific adaptation of the Hurtt et al. (2011) harmonized land use dataset. The historical to present-day transition highlights the global growth of croplands, from 5.3 million km$^2$ to 14.7 million km$^2$ at the expense of forests and grasslands. The net increase of 9.4 million km$^2$ of croplands matches values provided by Hurtt et al. (2011), indicating that the mapping of this dataset to the CLM PFTs preserved the change in cropland coverage. The CLM PFTs do not include a separate pasturelands category, therefore changes in
pasturelands (increase by 25.5 million km$^2$ from 1850 to 2000) in the Hurtt et al. (2011) dataset are mapped to grasslands in the CLM dataset. Figure 2 shows some regional increases in grassland coverage consistent with pasture expansion. Finally, we note that much of the agricultural expansion in Western Europe and Eastern North America pre-dates 1850, and thus a trend towards a return to forestlands is evident in these regions in Figure 2.

Figure 3 shows the change in leaf area index (LAI) associated with the historical to present-day change in land use.
Expansion of croplands leads to reductions in LAI, typically less than 20% locally. Globally, there is a 3% reduction in LAI

due to land use alone. We note that the feedback of increasing $CO_2$ fertilization on terrestrial productivity is not included here.

## 4 Impact of Historical Anthropogenic Land Use Change on Emissions and Deposition

Table 3 summarizes the changes in emissions driven by land use change (and associated agricultural activities) simulated in GEOS-Chem for the historical transition from 1850 to 2000. Global annual mean BVOC emissions of isoprene, monoterpenes, and sesquiterpenes decline by 10-12% due to the expansion of croplands (Figure 2) a vegetation class with very low basal emission rates for these BVOCs (Guenther et al., 2012). For example isoprene and α-pinene emission factors for croplands are at least 2 orders of magnitude less than for needleleaf or broadleaf trees. The distribution of these reductions is shown in Figure 4. Fractional declines are consistent year-round, with larger absolute decreases in summer at northern mid-latitudes following the seasonality of vegetation. These changes are more modest than the 35% decrease in global BVOC emissions due to land use change estimated by Unger (2014) over the same time period. Unger (2014) follows the same historical land use trajectory used here (Hurtt et al., 2011) however the GISS model mapping of this dataset includes pasturelands as part of the cultivation biome which also consists of croplands and does not emit BVOCs (personal communication, N. Unger). In contrast, the CLM approach maps pasturelands to grasslands, which are modest, but non-negligible, emitters of BVOCs. Therefore, the substantial difference between our estimate and that of Unger (2014) is associated with the uncertainty in characterizing BVOC basal emission rates from pasturelands, which expand significantly from 1850 to 2000. Ward et al. (2014) estimate only a 1% increase in all biogenic emissions due to historical LULCC, however, they do not disaggregate BVOCs and we cannot compare simulated changes in terpenes directly.

Figure 5 shows the estimated increases in nitrogen emissions associated with LUC. Global annual mean nitrogen oxide emissions from soils increase by 3.7 TgNyr$^{-1}$ (more than 50%) from 1850 to 2000. The majority of this increase (2.9 TgNyr$^{-1}$) is associated with enhanced fertilizer usage in 2000 compared to 1850, however emissions increase by 0.8 TgNyr$^{-1}$ due to shifts in biomes (and the associated emission factors) as well as increased escape of NOx from the canopy due to lower LAI in 2000 (Table 3). Relative changes in soil NOx emissions are consistent year-round. Heald and Spracklen (2015) estimated a 50% increase in soil NOx emissions associated with LUC, in good agreement with our estimate here, but to our knowledge no study has simulated the change in soil NOx emissions due to historical LUC. These results highlight the need to better constrain changes in soil NOx emissions due to fertilizer application over the last 150 years (Felix and Elliott, 2013). Figure 5 also shows that total ammonia emissions more than double from 1850 to 2000 due to agricultural sources, following the RCP emissions (van Vuuren et al., 2011). This reflects substantial increases in fertilizer usage on croplands and domesticated animals on pasturelands.

Historical LUC also modifies the surface properties that control the uptake of gases at the surface. This loss is most significant for tropospheric ozone, a relatively insoluble gas, which is biologically reactive, and is therefore readily taken up by vegetation (Stevenson et al., 2006; Wesely and Hicks, 2000). The response of ozone dry deposition velocity to changes in

land use is dominated by changes to surface resistance. Therefore changes to the aerodynamic resistances due to differences in roughness height (which increases from grassland to agriculture to forests, see Table A1 of Geddes et al. (2016)) do not substantially impact the simulated ozone dry deposition. Figure 6 shows that historical LUC has modestly increased O$_3$ deposition velocities over most regions where croplands have expanded. This increase is driven by lower stomatal and surface resistance values associated with croplands (compared to forests and grasslands) in the Wesely et al. (1989) scheme. This effect outweighs the decreases in deposition velocity associated with decreases in LAI over croplands (Figure 3). However, this is not the case in Southeast Asia, where replacement of dense tropical forests with croplands substantially decreases LAI (Figure 3), driving down deposition velocities. Local decreases in deposition velocity over Western Europe and the eastern United States are the result of reforestation of croplands since 1850. In southeastern Brazil, expansion of pasturelands (shown as grasslands in Figure 2) at the expense of broadleaf trees, leads to a decrease in deposition velocity. Local differences do not exceed 20% and are typically less than 10%. Historical LUC produces less than 1% difference in global mean ozone deposition velocity. Changes in deposition velocity shown in Figure 6 are relatively aseasonal, with somewhat larger changes in summer at northern mid-latitudes associated with peak vegetation density. Verbeke et al. (2015) explore the impact of future LUC in 2050 on the deposition of ozone. Qualitatively their simulated response to cropland expansion and reforestation are consistent with our results, with local changes to deposition velocities that are within 10%.

## 5 Impact of Historical Anthropogenic Land Use Change on Atmospheric Composition

The response of atmospheric composition to changes in biosphere-atmosphere fluxes depends on the assumed anthropogenic emissions; we first present results using present-day (2000) anthropogenic emissions, and comment below on differences when instead employing pre-industrial (1850) anthropogenic emissions (Tables 1 and 2).

Figures 7 and 8 show the impact of historical LUC on boreal summer (June-August) and winter (December-February) mean surface concentrations of key species. The decline in BVOC emissions driven by the expansion of croplands leads directly to widespread decreases in biogenic SOA (BSOA). Surface concentrations decrease by 14% on average; local BSOA concentrations in summertime decrease by up to 84% and increase by up to 54% over Western Europe and Eastern U.S., where BVOC emissions increase due to reforestation (see Figures 2, 3 and 4). The global annual mean tropospheric burden of BSOA decreases by 13% due to historical LUC (Table 4).

The more than doubling of ammonia emissions from pre-industrial conditions to present-day associated with agricultural activities (Table 3) dramatically enhances ammonium nitrate formation. This increase is particularly evident in northern mid-latitudes winter (Figure 8) where cool temperatures favour nitrate formation, and mean surface nitrate concentrations more than double. The global annual mean tropospheric burden of aerosol nitrate increases almost 4-fold due to historical LUC (Table 4). This increase is almost entirely the result of ammonia emissions increases; land use change alone (simulations 1 vs 2; see Tables 1 and 2) increases the tropospheric burden of nitrate by only 1.1%, stemming from the enhanced soil NOx

emissions. These results are consistent with Bauer et al. (2016) who estimate that agriculture is responsible for 78% of ammonia emissions, and that this is the prevailing source of ammonium nitrate formation in the Northern Hemisphere.

In summer, surface NOx concentrations are locally enhanced by LUC (Figure 7), driven by elevated soil NOx emissions. Despite this, we see that surface ozone concentrations decrease in the northern hemisphere. These decreases reflect elevated ozone deposition over croplands (Figure 6) and decreases in BVOC emissions (Figure 4). Summertime mean surface ozone decreases by up to 8.5 ppb, with at least a 1 ppb decrease throughout the Northern Hemisphere. The changes in emissions and uptake over the Southern Hemisphere lead to negligible changes in surface ozone (generally less than 1 ppb).

In winter, the large additional pool of atmospheric ammonia associated with anthropogenic LUC pulls nitric acid into the particle phase. As a result, nitric acid surface concentrations decrease by over 50% throughout the Northern Hemisphere (Figure 8). This reduces NOx recycling from nitric acid, leading to an overall decrease in NOx concentrations, despite increases in soil NOx emissions. Thus in winter, historical LUC has led to a drop in NOx and BVOC concentrations in the Northern Hemisphere, while ozone deposition velocities increase. Therefore, wintertime ozone decreases over northern mid-latitudes are of similar magnitude as in summer (up to 6.6 ppb, generally 1-2 ppb), despite reduced photochemical production of ozone, and thus lower absolute concentrations, in wintertime. Ozone changes in the Southern Hemisphere in winter remain small, but local increases of up to 2.5 ppb are simulated. In these NOx limited regions, increases in soil NOx emissions enhance ozone production and decreases in BVOC emissions reduce the sink of ozone via isoprene oxidation. In some regions, such as eastern Brazil, decreases in ozone deposition velocity due to expansion of pasturelands (Figure 6), bolster this enhancement.

While these changes in surface ozone concentrations are small, they are comparable to the so-called "climate penalty" increases in ozone associated with a warming climate (Tai et al., 2013; Wu et al., 2008). This suggests that both historical analyses and future projections of ozone air quality should consider land use conversion alongside emissions when characterizing the impacts of anthropogenic activities. Table 4 shows that the net annual mean tropospheric burden of ozone decreases only by 1.6% due to historical LUC, suggesting that ozone impacts on radiative forcing are considerably more modest. Global mean tropospheric OH changes by less than 0.5% due to historical LUC. Therefore in our simulations historical LUC has little impact on the tropospheric oxidative capacity or the tropospheric methane lifetime.

The above results characterize changes assuming that anthropogenic emissions are fixed at year 2000 levels. While it is necessary to fix anthropogenic emissions to isolate and quantify the effect of LUC, in reality, anthropogenic emissions and land use co-evolve. Thus, it is equally valid to assess the impact of LUC with simulations where anthropogenic emissions are fixed at 1850 fluxes (note that methane remains at year 2010 levels in these simulations). As shown in Table 2, we repeat all our simulations with these alternate anthropogenic emissions. Table 4 shows the global mean tropospheric burdens assessed under this scenario. Figures 9 and 10 can be compared to Figures 7 and 8 and show seasonal mean changes in surface concentrations when anthropogenic (non-agricultural) emissions are fixed at 1850 levels. Biogenic SOA burdens and concentrations are relatively unaffected by differences in anthropogenic emissions; very minor differences are associated with changes in oxidant levels. While the surface concentrations of NOx, $HNO_3$, and aerosol nitrate simulated under 1850

anthropogenic emissions are all considerably lower than estimated using year 2000 emissions, the qualitative patterns associated with LUC presented in Figures 9 and 10 are consistent, though more modest, than those presented in Figures 7 and 8. We see from Table 4, that with 1850 anthropogenic emissions, ammonium nitrate formation is NOx limited and a significant fraction of the ammonia emissions increase due to agricultural sources remains in the gas phase. Thus, the absolute increase in nitrate aerosol burden due to LUC is somewhat smaller (by 17%) than estimated using year 2000 anthropogenic emissions. As seen in Figures 9 and 10 surface concentrations are more sensitive to these effects with much smaller absolute concentration changes when using 1850 anthropogenic emissions (increases of less than 1 $\mu gm^{-3}$ in Figure 10 compared to wide-spread increases of more than 5 $\mu gm^{-3}$ when using 2000 anthropogenic emissions in Figure 8). This suggests that while ammonium nitrate formation is dramatically curtailed at the surface when anthropogenic NOx emissions are low, formation of ammonium nitrate in the free troposphere is not substantially impacted by reductions in anthropogenic NOx. Thus, the increase in surface nitrate from pre-industrial to present-day is controlled more by the rise in anthropogenic NOx emissions than the rise in agricultural ammonia emissions, while the increase in the burden of tropospheric nitrate is driven primarily by the increase in ammonia. Finally, while the change in global mean tropospheric burden of ozone is similar whether assuming 1850 or 2000 anthropogenic emissions, some spatial differences are apparent in surface concentrations. In particular, summertime surface $O_3$ concentrations locally increase (by up to 5 ppb) over Northern Hemisphere mid-latitudes regions (Figure 9) where soil NOx emissions increase due to LUC (Figure 5). Ozone production is widely NOx limited under 1850 anthropogenic emissions, and thus the ozone production efficiency of additional soil NOx emissions is considerably higher, and outweighs the impact of elevated deposition velocities for ozone due to LUC.

## 6 Direct Radiative Impacts of Historical Anthropogenic Land Use Change

The changes in annual mean tropospheric burden under 1850 and 2000 anthropogenic emissions shown in Table 4 bracket the potential impact of historical LUC on secondary PM and ozone. To estimate the change in direct radiative fluxes associated with historical LUC we apply monthly mean radiative efficiencies for BSOA, nitrate, and tropospheric ozone estimated from previous GEOS-Chem simulations (see Section 2) to our results using 2000 anthropogenic emissions. As this change is directly driven by anthropogenic LUC it represents the direct radiative forcing associated with land use change (LUC-DRF). Figure 11 summarizes these results.

The largest radiative impact from historical LUC in our simulations is a cooling of -0.071 $Wm^{-2}$ associated with the rise in nitrate aerosol from pre-industrial to present-day. This increase is driven almost entirely by increases in ammonia emissions. The LUC-DRF of nitrate constitutes 81% of the total direct radiative effect (DRE) of nitrate. Heald and Spracklen (2015) estimate a stronger LUC-cooling associated with nitrate (-0.094 $Wm^{-2}$), however this back-of-the-envelope calculation is based on a stronger overall radiative effect of nitrate.

We estimate that decreases in BSOA due to historical LUC have produced a warming of +0.017 $Wm^{-2}$. This LUC-DRF is ~10% of the DRE of BSOA in our simulations. This value is smaller than the LULCC change in DRE (+0.034 $Wm^{-2}$)

estimated by Heald and Spracklen (2015), however the later included $CO_2$ fertilization and inhibition effects and is therefore not directly comparable. The LUC-DRF of tropospheric ozone associated with the very small changes in global burden discussed in Section 5, is a cooling of -0.01 Wm$^{-2}$. Ward et al. (2014) estimate a LUC-DRF of opposite sign for ozone (+0.12 Wm$^{-2}$), however this value primarily reflects changes in methane and fire emissions, which we do not consider here; Ward et al. (2014) do not quantify the change in BVOC emissions. Our results are qualitatively consistent with the LUC-DRF of BSOA and tropospheric ozone estimated by Unger (2014), but are considerably more modest. This largely arises from the smaller change in BVOC emissions estimated in our study (~11%) compared to Unger (2014) (35%) due to different classifications of pasturelands and their associated BVOC emissions (see Section 3 and 4).

## 7 Conclusions

This study explores the simulated impact of historical LUC on air quality and DRF, with an emphasis on secondary formation of PM and tropospheric ozone. Land use change from pre-industrial (1850) to present-day (2000) is chiefly defined by cropland and pastureland expansion world-wide, as well as local reforestation in Western Europe and the eastern United States. This has led to a global decline in BVOC emissions (by ~11%), however the associated agricultural sources have increased emissions of both ammonia (by a factor of 2) and soil nitrogen oxides (by 50%). At the same time, surface uptake has been impacted by changes in vegetation type and density. Generally, we find that ozone deposition velocities have increased due to LUC, with some local exceptions associated with reforestation and expansion of pasturelands.

These LUC-driven changes in biosphere-atmosphere exchange processes work in concert to directly impact the secondary formation of BSOA, aerosol nitrate, and ozone. Surface air quality is significantly impacted by these changes, with a 14% average decrease in BSOA concentrations, a more than doubling of mean nitrate concentrations, and changes in surface $O_3$ of up to 8.5 ppb. We find that changes to ozone surface concentrations in the Northern Hemisphere are sensitive to the assumed anthropogenic emissions. This reflects the changing balance of deposition and precursor emissions of BVOCs and NOx in controlling ozone concentrations under varying NOx levels. Associated with these changes we estimate a DRF associated with LUC for nitrate (-0.071 Wm$^{-2}$), BSOA (+0.017 Wm$^{-2}$), and tropospheric ozone (-0.01 Wm$^{-2}$). We note that these estimates are obtained with fixed 2010 meteorology, and therefore we have not assessed the interannual climate variability against which these values can be compared for significance. While this is certainly not the first study to estimate the DRF of nitrate, few models have routinely assess this (Myhre et al., 2013), and to our knowledge this is the first that assesses the nitrate DRF associated with LUC. This study suggests that BSOA concentrations were elevated in the more extensively forested pre-industrial era. This higher pre-industrial burden of natural aerosol may temper the indirect aerosol effect (Carslaw et al., 2013; Menon et al., 2002), which we do not assess here. We attribute differences between our more modest estimates of LUC-DRF for BSOA and $O_3$ and those of Unger (2014) to differing treatments of pasturelands in the respective models, and thus the assumed BVOC basal emission rate for pasturelands. These substantial differences in LUC-

DRF highlight how uncertainty in the representation of historical land use change in earth system models leads to large uncertainties in global chemical composition.

This study examines only a subset of the emissions that may be impacted by LUC. In particular, we do not assess the changes in primary PM associated with LUC, including dust, smoke, and bioaerosol. It remains challenging to disaggregate the natural and anthropogenic influences on these emissions. In addition, we fix methane concentrations and therefore do not comprehensively assess how changes in global oxidative capacity driven by LUC may impact secondary aerosol and ozone formation. We also do not consider the meteorological feedbacks on atmospheric composition associated with land use change; more work is needed to quantify how these feedbacks compare to the direct perturbations associated with biosphere-atmosphere exchange. Thus, this study quantifies only part of the impacts of LUC. Furthermore, as our results rely heavily on the parameterization of biosphere-atmosphere exchange processes, more work is needed to validate these emissions and deposition schemes (e.g. (Hardacre et al., 2015)). In addition, given uncertainties in BSOA formation (Hallquist et al., 2009), and the general underestimate of OA in global models, including GEOS-Chem (Heald et al., 2011), the absolute magnitude of the impact of LUC on both air quality and DRF via BSOA may be underestimated here. Finally, uncertainties associated with the gas-phase oxidation chemistry of isoprene, monoterpenes, and sesquiterpenes may impact our simulated sensitivity of BSOA and $O_3$ to LUC. The simulations analysed in this study were performed with one chemical transport model (GEOS-Chem); the degree to which model-specific treatments of chemical oxidation, aerosol formation, emissions, removal, and meteorology may impact the results cannot be assessed here. Thus, additional modelling investigations using alternate model schemes are required to better characterize the uncertainty surrounding the impact of land use change on air quality and climate forcing.

We find that historical land use change has brought about substantial changes in secondary PM and ozone formation, impacting air quality and direct radiative forcing. The magnitude of these changes are comparable to the feedbacks associated with climate change (Tai et al., 2012; Tai et al., 2013). Furthermore, in an era of declining emissions of air pollution precursors (Smith and Bond, 2013), anthropogenic land use change may become the dominant human impact on atmospheric composition. Therefore, more work is needed to improve our understanding and parameterization of biosphere-atmosphere exchange processes, and how these are altered by changing vegetation.

**Acknowledgements**

This work was supported by NSF (ATM-0929282 and ATM-1564495). JAG acknowledges support from an NSERC CREATE IACPES postdoctoral fellowship and travel grant.

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

**Tables**

Table 1: List of GEOS-Chem simulations with relevant input parameters

| | Simulation Name | Land Use | Anthropogenic (non-Agricultural) Emissions | Agricultural Emissions |
|---|---|---|---|---|
| 1 | 2000L2000E | 2000 | 2000 | 2000 |
| 2 | 1850L2000E | 1850 | 2000 | 2000 |
| 3 | 2000L1850E | 2000 | 1850 | 1850 |
| 4 | 1850L1850E | 1850 | 1850 | 1850 |
| 5 | 2000L2000E1850NH3 | 2000 | 2000 | 1850 |
| 6 | 2000L1850E2000NH3 | 2000 | 1850 | 2000 |
| 7 | 1850L2000E1850NH3 | 1850 | 2000 | 1850 |

5    Table 2: Roadmap for how simulations are combined to estimate the impact of land use change and the associated change in agricultural emissions on air quality

| | 2000 Anthropogenic Emissions | 1850 Anthropogenic Emissions |
|---|---|---|
| Land Use Change Alone | 1-2 | 3-4 |
| Agricultural Emissions Alone | 1-5 | 6-3 |
| Land Use Change + Agricultural Emissions | 1-7 | 6-4 |

Table 3: Annual average emissions impacted by historical land use change alone, shown separately are changes in emissions due to both land use change and associated agricultural emissions.

| | Land use change alone | | | Land use change + associated agricultural emissions | | |
|---|---|---|---|---|---|---|
| | 1850 | 2000 | % change | 1850 | 2000 | % change |
| Isoprene (Tgyr$^{-1}$) | 518 | 459 | -11.4% | 518 | 459 | -11.4% |
| Monoterpenes (Tgyr$^{-1}$) | 188 | 165 | -12.0% | 188 | 165 | -12.0% |
| Sesquiterpenes (Tgyr$^{-1}$) | 23.8 | 21.3 | -10.6% | 23.8 | 21.3 | -10.6% |
| Ammonia (Tgyr$^{-1}$) | 59.3 | 59.3 | 0.0% | 28.4 | 59.3 | +109% |

| Soil NOx (TgNyr$^{-1}$)[1] | 9.2 | 10.0 | +8.4% | 6.3 | 10.0 | +58.5% |

---

[1] Soil NOx emissions tabulated here are when non-agricultural anthropogenic emissions are held at year 2000 levels (simulations 1 and 7). Reduced anthropogenic emissions in 1850 lower soil NOx re-emissions levels slightly (but totals are within 2%)

**Table 4**: Annual average tropospheric burden (Tg) of key species. Also shown is the changes driven by historical land use change (including associated agricultural emissions). Values estimated using year 2000 anthropogenic emissions and 1850 anthropogenic emissions are shown.

| | Year 2000 Anthropogenic Emissions | | | Year 1850 Anthropogenic Emissions | | |
|---|---|---|---|---|---|---|
| | 1850 | 2000 | 2000-1850 (%) | 1850 | 2000 | 2000-1850 (%) |
| Biogenic SOA | 0.59 | 0.52 | -0.076 (-13%) | 0.58 | 0.50 | -0.076 (-13%) |
| $O_3$ | 266 | 262 | -4.2 (-1.6%) | 231 | 228 | -3.1 (-1.4%) |
| Ammonia | 0.05 | 0.14 | +0.09 (+190%) | 0.09 | 0.28 | +0.19 (+219%) |
| Nitrate | 0.07 | 0.31 | +0.25 (+374%) | 0.07 | 0.27 | +0.20 (+305%) |

**Figures**

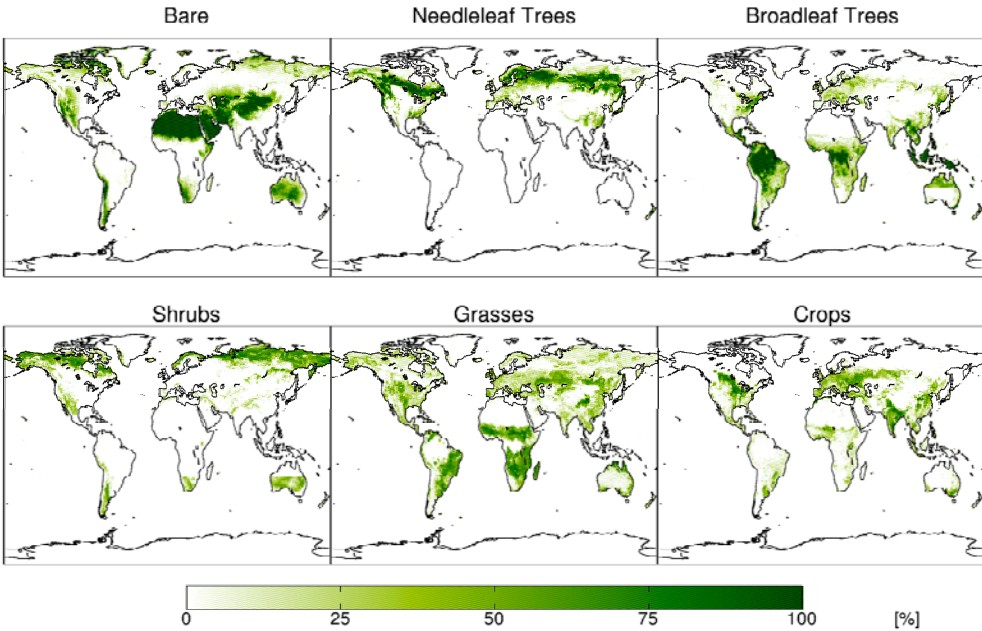

**Figure 1: Present day (year 2000) percentage of land area occupied by six classes of vegetation.**

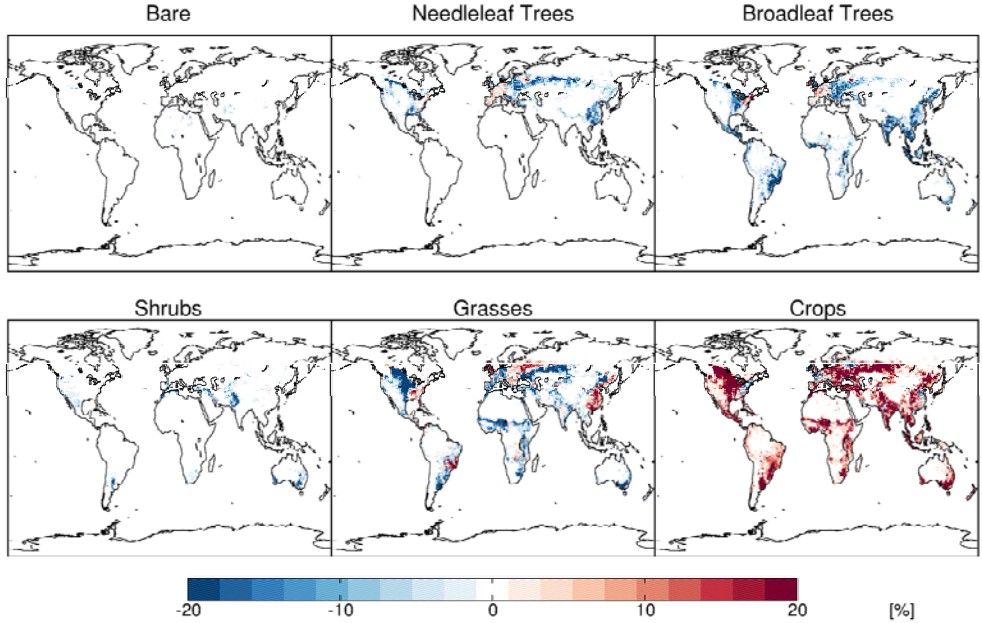

5  **Figure 2: Change from pre-industrial (1850) to present-day (2000) in the percentage of land area occupied by six classes of vegetation.**

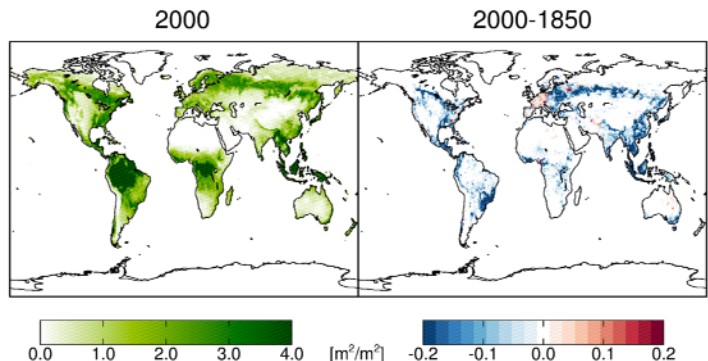

**Figure 3: Annual average leaf area index (LAI) in present-day (left) and the change in LAI from pre-industrial (1850) to present-day (2000) (right).**

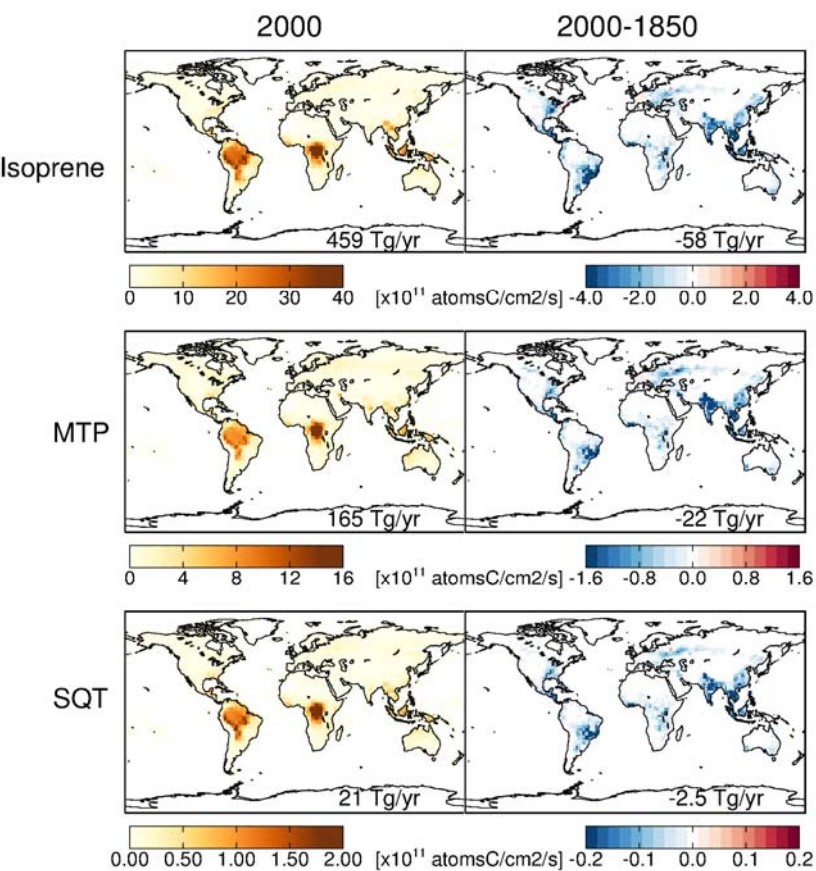

5  **Figure 4: Annual mean simulated emissions of BVOCs from vegetation. Total emissions for present-day (2000) shown on the left; the change due to historical land use change is shown on the right. Global annual emission values are shown inset.**

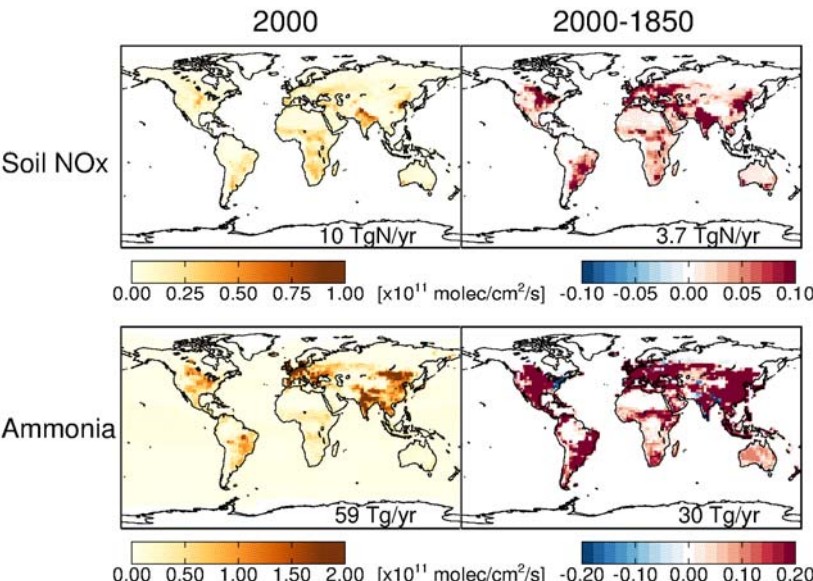

**Figure 5: Annual mean emissions of nitrogen oxides from soils (top row) and ammonia (bottom). Total emissions for present-day (2000) shown on the left; the change due to historical land use change (and the associated agricultural emissions) is shown on the right. Global annual emission values are shown inset.**

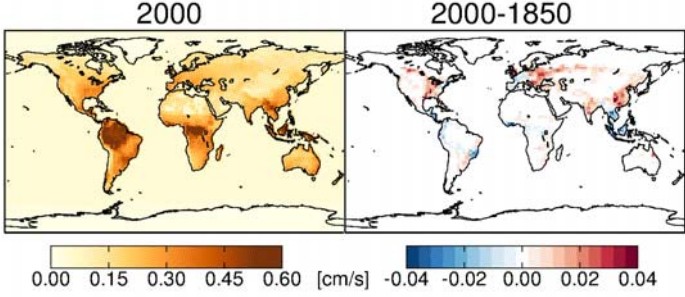

**Figure 6: Annual mean simulated dry deposition velocity of ozone for present-day (2000) shown on the left; the change due to historical land use change is shown on the right.**

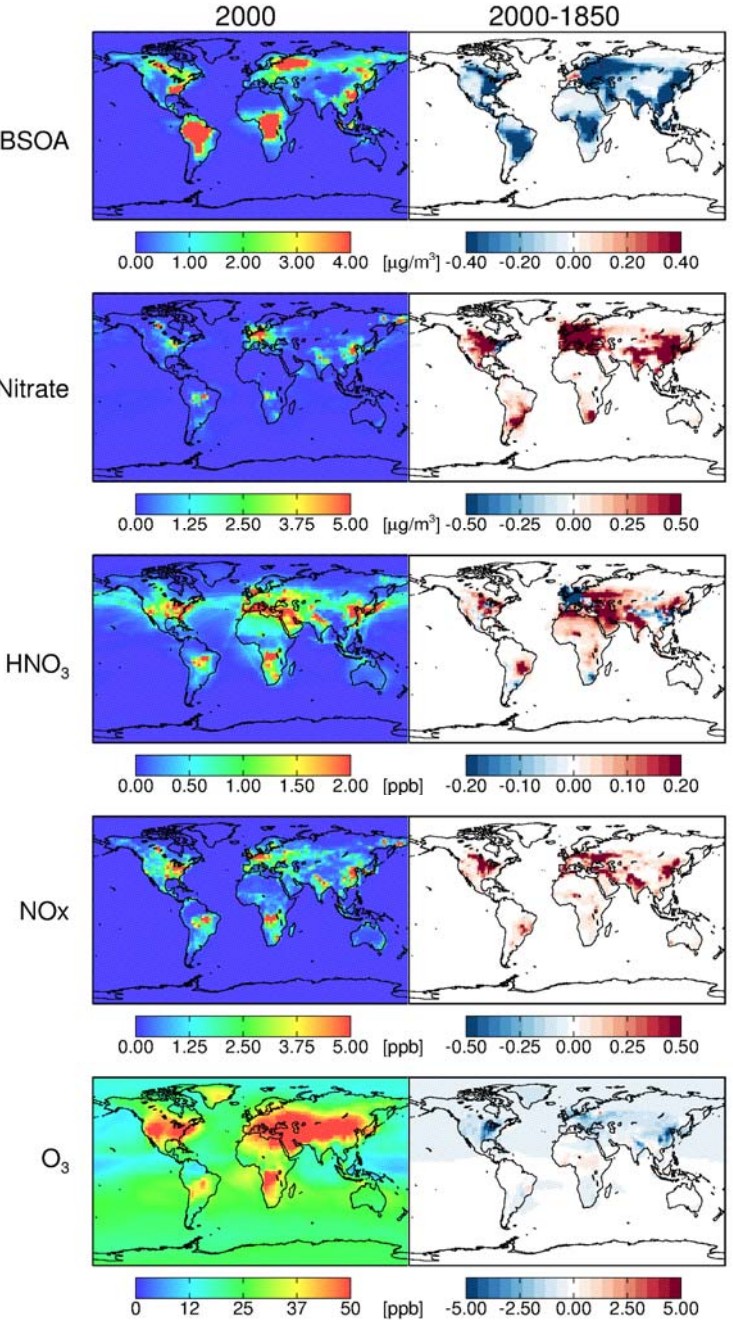

**Figure 7: Boreal summertime (June-August) mean simulated surface concentrations of biogenic SOA (BSOA), aerosol nitrate, nitric acid (HNO₃), nitrogen oxides (NOx), and ozone. Concentrations for present-day (2000) shown on the left; the change due to historical land use change is shown on the right. All simulations performed with**

present-day (2000) anthropogenic emissions; shown here are the differences between simulations 1 and 7 (see Tables 1 and 2).

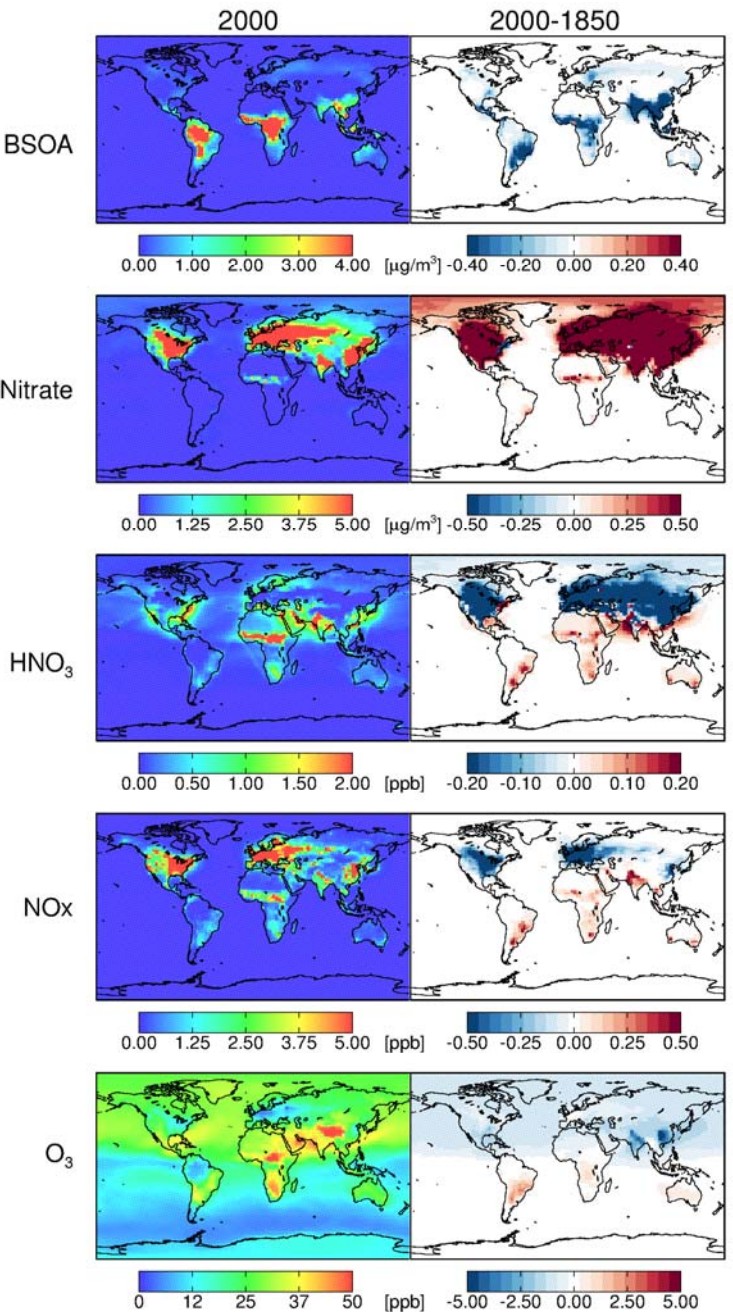

**Figure 8: Boreal wintertime (December-February) mean simulated surface concentrations of biogenic SOA (BSOA), aerosol nitrate, nitric acid (HNO₃), nitrogen oxides (NOx), and ozone. Concentrations for present-day (2000) shown**

on the left; the change due to historical land use change is shown on the right. All simulations performed with present-day (2000) anthropogenic emissions; shown here are the differences between simulations 1 and 7 (see Tables 1 and 2).

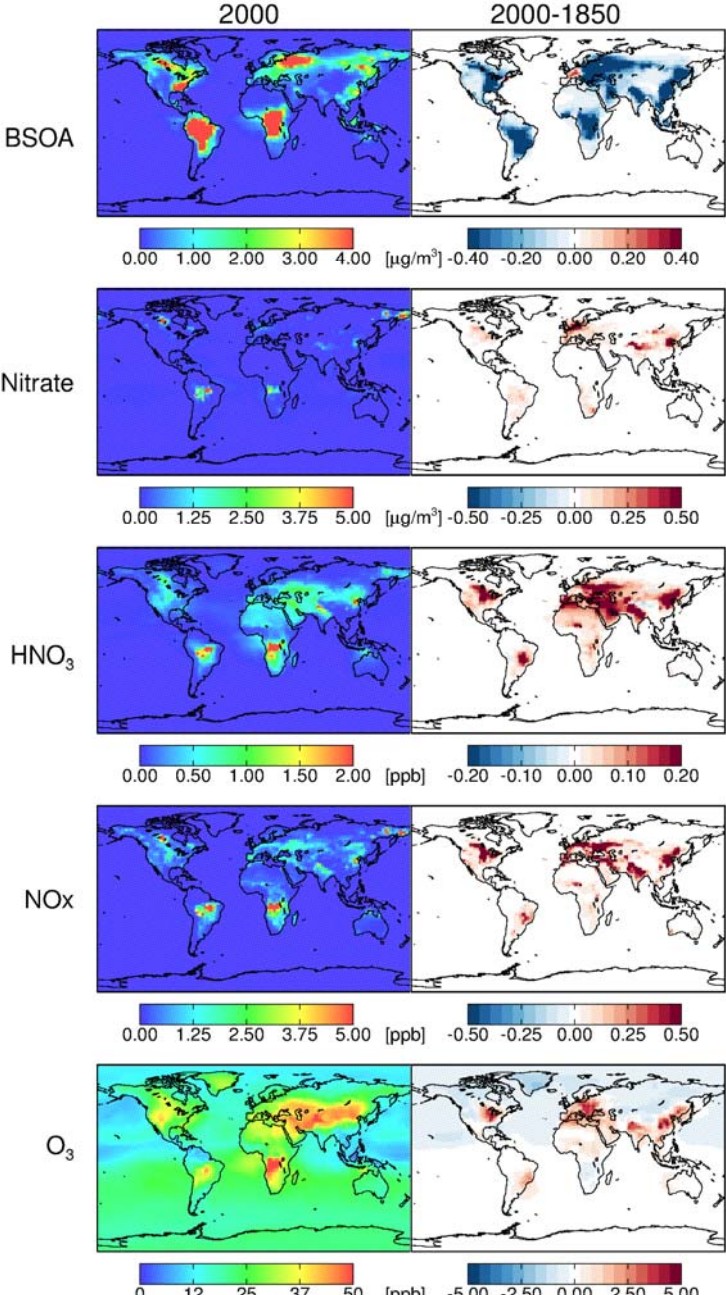

**Figure 9: Boreal summertime (June-August) mean simulated surface concentrations of biogenic SOA (BSOA), aerosol nitrate, nitric acid (HNO₃), nitrogen oxides (NOx), and ozone. Concentrations for present-day (2000) shown on the left; the change due to historical land use change is shown on the right. All simulations performed with pre-industrial (1850) anthropogenic emissions; shown here are the differences between simulations 6 and 4 (see Tables 1 and 2). Shown with same color bars as Figure 7 for comparison.**

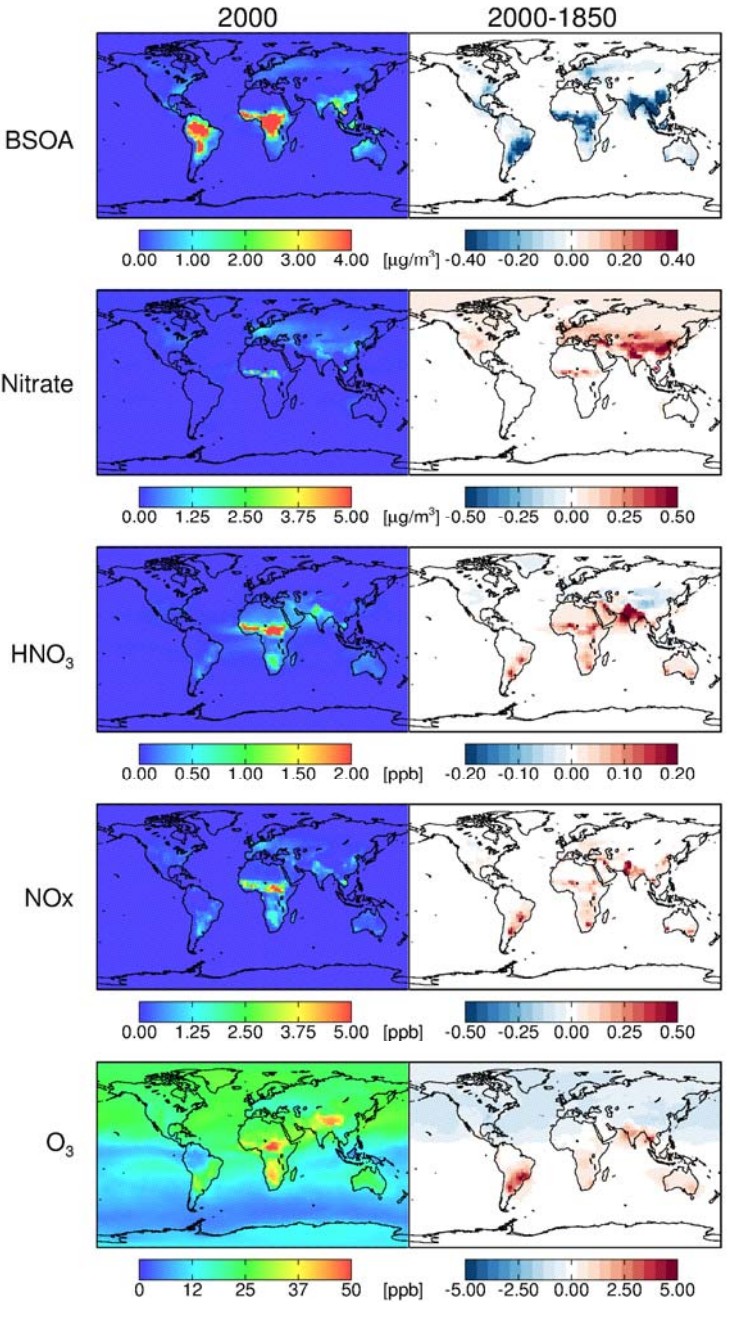

**Figure 10: Boreal wintertime (December-February) mean simulated surface concentrations of biogenic SOA (BSOA), aerosol nitrate, nitric acid (HNO₃), nitrogen oxides (NOx), and ozone. Concentrations for present-day (2000) shown on the left; the change due to historical land use change is shown on the right. All simulations performed with pre-industrial (1850) anthropogenic emissions; shown here are the differences between simulations 6 and 4 (see Tables 1**

5     **and 2). Shown with same color bars as Figure 8 for comparison.**

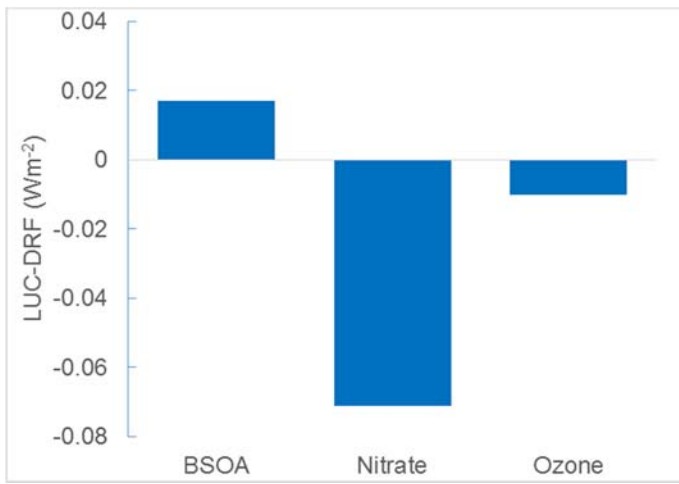

**Figure 11: Global annual mean direct radiative forcing associated with anthropogenic land use change (LUC-DRF)**

10     **from 1850 to 2000.**