# Peer review of "The Impact of Historical Land Use Change From 1850 to 2000 on Secondary Particulate Matter and Ozone"

_Atmospheric Chemistry and Physics, 2016_

## Referee Comment (RC1) · Anonymous Referee #1 · 6 Oct 2016

This fascinating paper applies a systems-engineering modeling approach to assess the impact of 1850 versus 2000 land cover fraction changes on short-lived precursor emissions and the resultant atmospheric ozone and aerosol effects. The study examines the chemical sensitivity to the land cover change fractions under both 1850 and 2000 chemical background states (in terms of anthropogenic short-lived precursor emissions not the physical climate), in part to offer a useful uncertainty range estimate. The NH3/NOx/nitrate results are especially interesting and consistent with the recent Bauer et al., GRL 2015 results, also the Lelieveld et al., Nature, 2015 finding that agriculture is the main driver of particulate-related human health impacts in Europe. The paper is an important contribution to emerging land-chemistry-climate science and de-

serves to be published in ACP once the following technical issues are addressed:

1. The main concern is that all simulations apply year 2010 meteorology. The underlying assumption is that surface albedo, energy and water changes caused by different land cover types have zero impact on atmospheric chemistry and aerosols. The manuscript needs to include clear statements about the missing meteorological feedbacks. Perhaps a more appropriate title would be something like: "..Impact of Historical Land Use Change Emissions. . .."

2. There is an extensive existing literature and multiple international assessment programs on the climatic effects of human land cover change through biophysical, albedo and meteorological changes. This paper essentially has pre-concluded that these water/energy/radiative changes have negligible impact on chemistry and aerosols, and only the short-lived precursor emissions changes are important. In fact, the meteorological/biophysical effects are apparently so unimportant to chemistry that they are not even mentioned. What is needed is a clear discussion of why albedo/biophysical/meteorological feedbacks have not been included in this analysis and how their inclusion would impact the results.

3. It is an excellent and efficient strategy to archive gridded radiative efficiencies that can be used in conjunction with global CTM-derived ozone and aerosol burden changes to assess radiative forcing impacts. However, the radiative efficiencies applied in this study are based on present day surface albedo and atmospheric water and cloud content etc. This seems to be wrong. The scattering aerosols (nitrate and BSOA) are quite sensitive to underlying surface albedo. For example, whether the underlying surface is covered in dark forest or bright crop/pasture can have a large quantitative impact on the local/regional aerosol radiative forcing. Atmospheric water content and cloud cover will also impact the aerosol radiative forcing results. There may be some effects on the SW ozone forcing too.

4. Assessing the impacts of land cover change on climate is a challenging multidisciplinary field. Therefore, it is critical for the chemistry-climate community to be extremely clear in their novel assessments that may be viewed by scientists from other disciplines (e.g. carbon cycle, surface biophysical climate communities) who may be less familiar with nuances in the atmospheric chemistry. For this reason, I recommend to modify the paper text carefully such that forcing values from Ward et al., 2014, Unger et al., 2014 and this study are not directly compared. These 3 exciting studies all examine rather different aspects of human land cover change effects on chemistry-climate inter-actions using completely different experimental design protocols. Indeed, the authors are quick to point out when their own results differ: "This value is smaller than the LULCC change in DRE (+0.034 Wm-2) estimated by Heald and Spracklen (2015). . ..is therefore not directly comparable". Yet, they proceed to compare Ward et al. 2014 and Unger et al. 2014: "We note here that the forcing estimated by Unger (2014) is of the opposite sign of that estimated by Ward et al. (2014)", which is fundamentally mislead-ing to readers. Ward et al., 2014 includes methane, dust, fire and carbon changes, whereas Unger (2014) focuses on BVOC emission changes and physical climate feed-backs. Consider that global chemistry-climate models give a wide spread of ozone and aerosol results even when based on carefully designed harmonized experimental protocols (e.g. ACC-MIP)! Here, the authors are attempting to compare directly quan-titative values from totally un-harmonized experiments that address different forcing components. Ultimately, a coordinated LU-AerChem-MIP multi-model assessment is needed if a single attributed LULCC-chemical forcing of global climate is to be deter-mined, (if that exists).

5. Methane. The authors include an interesting discussion of the oxidation capac-ity consequences of omitting the expanding rice production. In addition, (i) animal husbandry/livestock is a large global source of methane emissions (ii) the land use-induced short-lived precursor emission changes (NOx, BVOCs etc.) would influence the methane lifetime itself too.

6. The authors conclude that the BSOA and ozone global forcing results are qualitatively similar to those in Unger et al., 2014, but that the magnitude difference is mostly caused by differences in estimating the BVOC emissions change due to land use change. Does this mean that global BVOC-ozone and BSOA land use forcing depends mostly on BVOC emissions and is largely independent of the (complex) BVOC photochemical oxidation mechanisms under development? i.e. are current massive uncertainties in isoprene oxidation under different NOx regimes largely irrelevant for BVOC impacts on global climate? Based on current available information/evidence, the answer is yes.

7. In addition to the mapping of human land use onto the x-PFTs in global models, I suggest that the assigned PFT-specific basal emission rate for BVOCs is a large driver of the uncertainty too that needs to be discussed. For example, at least, if this study assigned a zero or v. low basal rate for pasture/grass PFT, would the results be even more consistent with Unger et al., 2014?

8. Does the SOA condensation model here depend on pre-existing OA levels? In which case the BSOA results would be sensitive to the assumed fire emissions that are prescribed to year 2010 in the study. Would burning be higher in some regions in 1850, leading to potentially even higher PI BSOA than reported here?

9. For IPCC-standard radiative forcing results, uncertainty ranges are needed. Naked values like -0.071 Wm-2, and -0.01 Wm-2 seem small and meaningless esp. without uncertainty ranges. Are these numbers statistically significant with 95% confidence relative to inter-annual climate variability in the model? Quantitative information is needed on the statistical robustness of the results. The authors argue in the abstract that these global forcing values are 'substantial'. That is a matter of debate e.g. the CO2 value is 1.8 Wm-2. Are 0.5-2% of the CO2 historical forcing values 'substantial'? Or are they simply lost in inter-annual climate variability?

---

## Referee Comment (RC2) · Anonymous Referee #2 · 13 Oct 2016

Heald and Geddes "The Impact of Historical Land Use Change From 1850 to 2000 on Particulate Matter and Ozone"

GENERAL

This paper examined the impacts of historical land use change (LUC) and the associated agricultural emission change (AEC) on ozone and secondary particulate matter between preindustrial and present day. The main conclusion is that LUC+AEC result in increased burden of nitrate but decreased burden of BSOA and ozone. Such changes further induce radiative perturbations which present a strong cooling forcing since 1850. This is a fantastic work and analyses are comprehensive. Some minor revisions are required before the publication.

1. Some results presented in the study may be model dependent. The authors applied GEOS-Chem (GC) model in their study. Although the GC is a widely used and validated CTM, some inherent characteristics may definitely affect the changes in atmospheric chemistry. For example, to explain why the surface nitrate shows large deviations but tropospheric nitrate burden shows small differences between simulations using 1850 and 2000 anthropogenic emissions, the authors claim that "the increase in surface nitrate from pre-industrial to present-day is controlled more by the rise in anthropogenic NOx emissions than the rise in agricultural ammonia emissions, while the increase in the burden of tropospheric nitrate is driven primarily by the increase in ammonia". Are there any observations supporting such conclusion? Similar problems exist for ozone changes (shown in the detailed comments below). The authors need to discuss the possible uncertainties of these responses and reminder readers that the predicted changes in atmospheric composition is somewhat model-dependent.

2. The authors performed sensitivity experiments to isolate the impacts of LUE and AEC (Table 2) but did not present those results in their analyses. Based on the qualitative explanations, we can understand that the large enhancement of nitrate is mainly attributed to AEC, the reductions in biogenic secondary organic aerosols (BSOA) is dominantly driven by LUC, and the decline of ozone burden is a compound result of AEC and LUC, and the impacts of LUC seem to overweigh that of AEC. However, without quantitative numbers, we do not know the individual contributions of LUC and AEC. I suggest that the authors add a new Table to summarize changes in atmospheric composition due to different drivers (LUC, AEC, and LUC+AEC) as indicated in Table 2.

3. Definition of LUC is confusing. Sometimes, LUC refers to LUC+AEC: "The global annual mean tropospheric burden of aerosol nitrate increases almost 4-fold due to historical LUC (Table 4)". In the following sentence, however, LUC refers to land use change alone: "This increase is almost entirely the result of ammonia emissions increases; land use change alone (simulations 1 vs 2; see Tables 1 and 2) increases the

tropospheric burden of nitrate by only 1.1%". In addition, the phrase "land use change" is used frequently after the definition of abbreviation "LUC" in the paper. Similar problem exists for 'DRF' and 'BSOA'. Some clean-up work is required for the clarity.

SPECIFIC

1. The title of the paper may be more appropriate as "The Impact of Historical Land Use Change From 1850 to 2000 on Ozone and Secondary Particulate Matter"

2. Page 9 Line 2: "where soil NOx emissions increase due to land use change", here NOx emissions are due to AEC instead of LUC. Similar statement in the paper needs to be clarified.

3. Page 9 Lines 2-5: "Ozone production is widely NOx limited under 1850 anthropogenic emissions, and thus the ozone production efficiency of additional soil NOx emissions is considerably higher, and outweighs the impact of elevated deposition velocities for ozone due to LUC" This cannot explain why the burden of ozone is still decreased due to LUC with 1850 anthropogenic emissions.

4. Page 9 Line 19: "DRE" means "direct radiative effect" or just typo for "DRF"?

5. Figure 5 caption: Changes of soil NOx and ammonia are caused by AEC instead of LUC.
* * *

---

## Author Comment (AC1) · 15 Nov 2016

**Response to Reviews**

We would like to thank the reviewers for their comments and suggestions. We have addressed these below (in blue below original comment).

**REVIEWER #1**

This fascinating paper applies a systems-engineering modeling approach to assess the impact of 1850 versus 2000 land cover fraction changes on short-lived precursor emissions and the resultant atmospheric ozone and aerosol effects. The study examines the chemical sensitivity to the land cover change fractions under both 1850 and 2000 chemical background states (in terms of anthropogenic short-lived precursor emissions not the physical climate), in part to offer a useful uncertainty range estimate. The NH3/NOx/nitrate results are especially interesting and consistent with the recent Bauer et al., GRL 2015 results, also the Lelieveld et al., Nature, 2015 finding that agriculture is the main driver of particulate-related human health impacts in Europe. The paper is an important contribution to emerging land-chemistry-climate science and deserves to be published in ACP once the following technical issues are addressed:

We appreciate these supportive comments and are pleased that the article was well received.

1. The main concern is that all simulations apply year 2010 meteorology. The underlying assumption is that surface albedo, energy and water changes caused by different land cover types have zero impact on atmospheric chemistry and aerosols. The manuscript needs to include clear statements about the missing meteorological feed backs. Perhaps a more appropriate title would be something like: "..Impact of Historical Land Use Change Emissions"

We agree with the reviewer that (by design) we did not consider how feedbacks from meteorological changes (driven by land use change) could impact atmospheric composition. The goal of the paper (to focus on biosphere-atmosphere exchange processes only) is provided in the first paragraph, however, we agree with the reviewer that we should better clarify that we are excluding meteorological feedbacks in our study (see text additions below). Given that we explored the impact of land use change on deposition as well as emissions, the suggested modification to the title would not be appropriate.

Modifications:
    a. Page 3, line 12-15: edited sentence (additions in bold): "we aim to complement previous investigations and explore the impacts of historical global anthropogenic land use change on **biosphere-atmosphere exchange processes and the resulting perturbations to secondary** PM and ozone."
    b. Page 4, lines 25-29: added text: "Land use change modulates surface albedo, energy, and water exchange (Pielke et al., 2002; Pielke et al., 2011; Pitman et al., 2009) which may feedback on atmospheric composition (Ganzeveld et al., 2010; Ganzeveld and Lelieveld, 2004). Unger (2014) suggest that these feedbacks are small compared to the perturbation in BVOC emissions from historical land use change. By design, by fixing meteorology at year 2010, we do not quantify these impacts in this study. Rather, our simulations focus on the direct impact of changes in biosphere-atmosphere exchange."

2. There is an extensive existing literature and multiple international assessment programs on the climatic effects of human land cover change through biophysical, albedo and meteorological changes. This paper essentially has pre-concluded that these water/energy/radiative changes have negligible impact on chemistry and aerosols, and only the short-lived precursor emissions changes are important. In fact, the meteorological/biophysical effects are apparently so unimportant to chemistry that they are not even mentioned. What is needed is a clear discussion of why albedo/biophysical/meteorological feedbacks have not been included in this analysis and how their inclusion would impact the results.

The reviewer is correct, and we certainly did not mean to give the impression that this body of work does not exist or that these impacts are not important. Rather, these processes are not the focus of our analysis. We believe that the text addition described in the point above clarifies this. We have also added some text in the conclusions to reiterate this point.

Modifications:
  c. See above point b
  d. Page 11, lines 11-13: added text: "We also do not consider the meteorological feedbacks on atmospheric composition associated with land use change; more work is needed to quantify how these feedbacks compare to the direct perturbations associated with biosphere-atmosphere exchange."

3. It is an excellent and efficient strategy to archive gridded radiative efficiencies that can be used in conjunction with global CTM-derived ozone and aerosol burden changes to assess radiative forcing impacts. However, the radiative efficiencies applied in this study are based on present day surface albedo and atmospheric water and cloud content etc. This seems to be wrong. The scattering aerosols (nitrate and BSOA) are quite sensitive to underlying surface albedo. For example, whether the underlying surface is covered in dark forest or bright crop/pasture c*a*n have a large quantitative impact on the local/regional aerosol radiative forcing. Atmospheric water content and cloud cover will also impact the aerosol radiative forcing results. There may be some effects on the SW ozone forcing too.

The reviewer is correct that we fixed surface properties to present-day values. This may be a limitation of our study (the use of present-day clouds and water vapor are consistent with our use of present-day meteorology) and we have now added text to clarify this assumption. However, for scattering aerosols we do not expect the impact of modest changes in surface albedo (the Encyclopedia of Soil Science gives the range of albedo for forests as 0.05-0.2 and for croplands as 0.1-0.25) on TOA radiative fluxes to be large (see for example, Haywood and Shine, 1997).

Modification:
  e. Page 4, line 20-21: added text: "We note that these radiative efficiencies are estimated using present-day land reflectances."

4. Assessing the impacts of land cover change on climate is a challenging multidisciplinary field. Therefore, it is critical for the chemistry-climate community to be extremely clear in their novel assessments that may be viewed by scientists from other disciplines (e.g. carbon cycle, surface biophysical climate communities) who may be less familiar with nuances in the atmospheric chemistry. For this reason, I recommend to modify the paper text carefully such that forcing values from Ward et al., 2014, Unger et al., 2014 and this study are not directly compared. These 3 exciting studies all examine

rather different aspects of human land cover change effects on chemistry-climate interactions using completely different experimental design protocols. Indeed, the authors are quick to point out when their own results differ: "This value is smaller than the LULCC change in DRE (+0.034 Wm-2) estimated by Heald and Spracklen (2015)...is therefore not directly comparable". Yet, they proceed to compare Ward et al. 2014 and Unger et al. 2014: "We note here that the forcing estimated by Unger (2014) is of the opposite sign of that estimated by Ward et al. (2014)", which is fundamentally misleading to readers. Ward et al., 2014 includes methane, dust, fire and carbon changes, whereas Unger (2014) focuses on BVOC emission changes and physical climate feedbacks. Consider that global chemistry-climate models give a wide spread of ozone and aerosol results even when based on carefully designed harmonized experimental protocols (e.g. ACC-MIP)! Here, the authors are attempting to compare directly quantitative values from totally un-harmonized experiments that address different forcing components. Ultimately, a coordinated LU-AerChem-MIP multi-model assessment is needed if a single attributed LULCC-chemical forcing of global climate is to be determined, (if that exists).

We appreciate the reviewer's point. And yet, given that there are limited studies on this topic, we think it is important that the quantitative results (and how they differ) be clearly discussed. Our objective in contrasting the results of Ward et al. and Unger in the Introduction was to highlight that they reach very different overall conclusions regarding the impact on land use change. The text does carefully indicate how these studies differ so the reader is made aware that they did not assess the same changes in the same way. To emphasize this we have added an additional sentence to the Introduction.

Modification:
   f.  Page 3, lines 7-8: added text "However, it is critical to note that these studies differ fundamentally in design and in the processes and species considered, highlighting the complexity of this forcing and the need to quantify specific impacts."

5.  Methane. The authors include an interesting discussion of the oxidation capacity consequences of omitting the expanding rice production. In addition, (i) animal husbandry/livestock is a large global source of methane emissions (ii) the land use-induced short-lived precursor emission changes (NOx, BVOCs etc.) would influence the methane lifetime itself too.

We agree that livestock represents an additional source perturbation associated with land use conversion. We note that we have assessed the impact of short-lived precursor emission changes on OH (and the methane lifetime) in Section 5, but this does not feedback on methane concentrations. We add these points to the text.

Modifications:
   g.  Page 4, lines 31: text added in bold: "changes in local methane sources (e.g. expansion of rice paddies, **growth in livestock**)."
   h.  Page 4, lines 31-34: added text: "Methane concentrations also do not respond to the changes in oxidative capacity associated with land-use driven changes in short-lived precursor emissions (assessed in Section 5)."

6.  The authors conclude that the BSOA and ozone global forcing results are qualitatively similar to those in Unger et al., 2014, but that the magnitude difference is mostly caused by differences in estimating the BVOC emissions change due to land use change. Does this mean that global BVOC-ozone

and BSOA land use forcing depends mostly on BVOC emissions and is largely independent of the (complex) BVOC photochemical oxidation mechanisms under development? i.e. are current massive uncertainties in isoprene oxidation under different NOx regimes largely irrelevant for BVOC impacts on global climate? Based on current available information/evidence, the answer is yes.

This is a great question! We have not compared the chemical mechanism in the GISS model used by Unger and the GEOS-Chem simulation we used, however it is likely that there are significant differences in the treatment of BVOC oxidation, and we agree that uncertainties on this are large (we state this as a source of uncertainty in our results in the Conclusions). However, it does appear that to first order, the different treatment of BVOC emissions is the primary factor responsible for the differences in our study (as stated in the text). Quantifying the relative role of chemistry and emissions is beyond our capabilities (we do not have access to the GISS model) or the scope of this paper, but it would indeed be an interesting question to address in a LUC-MIP. In addressing another point by Reviewer #2 (Modification k listed below) we have added text to the Conclusions calling for a LUC-MIP.

7. In addition to the mapping of human land use onto the x-PFTs in global models, I suggest that the assigned PFT-specific basal emission rate for BVOCs is a large driver of the uncertainty too that needs to be discussed. For example, at least, if this study assigned a zero or v. low basal rate for pasture/grass PFT, would the results be even more consistent with Unger et al., 2014?

Yes, the Reviewer is correct. This was implicit in our description of the differences between our treatment and Unger and stated on page 6. We have added additional text to re-iterate this point in the conclusions.

Modification:
   i.   Page 11 lines 4-5: text added in bold: "We attribute differences between our more modest estimates of LUC-DRF for BSOA and O3 and those of Unger (2014) to differing treatments of pasturelands in the respective models, **and thus the assumed BVOC basal emission rate for pasturelands**."

8. Does the SOA condensation model here depend on pre-existing OA levels? In which case the BSOA results would be sensitive to the assumed fire emissions that are prescribed to year 2010 in the study. Would burning be higher in some regions in 1850, leading to potentially even higher PI BSOA than reported here?

Yes, the model includes a reversible partitioning scheme which is dependent on pre-existing OA levels. Some studies indicate that fire activity was higher in pre-industrial and has declined due to the influence of human suppression (e.g. Marlon et al., 2008; Kloster et al., 2010), though some inventories suggest that fire emissions were lower in 1850 (e.g. Lamarque et al. 2010). Regardless, we are interested only in characterizing the changes in SOA driven by LUC, therefore any additional changes in OA partitioning driven by changes in fire emissions should be attributed as a fire feedback (not a LUC-driven effect). As we state on page 5, agricultural fires associated with cleared land make up a very modest fraction of total global fire emissions, therefore accounting for the LUC-driven fire emissions would modestly (likely negligibly) impact the global BSOA burden. Active deforestation could of course dramatically impact OA partitioning in a given year, but this is not a long-term perturbation, and is therefore not characterized here.

9. For IPCC-standard radiative forcing results, uncertainty ranges are needed. Naked values like -0.071 Wm-2, and -0.01 Wm-2 seem small and meaningless esp. without uncertainty ranges. Are these numbers statistically significant with 95% confidence relative to inter-annual climate variability in the model? Quantitative information is needed on the statistical robustness of the results. The authors argue in the abstract that these global forcing values are 'substantial'. That is a matter of debate e.g. the $CO_2$ value is 1.8 Wm-2. Are 0.5-2% of the $CO_2$ historical forcing values 'substantial'? Or are they simply lost in inter-annual climate variability?

We agree that uncertainties are required for an IPCC-type of assessment, however as these experiments are not representative of the type of chemistry-climate experiment included in the IPCC, we have neither the ensemble nor the multi-model statistics that enable this kind of estimate. And as we do not characterize any climate feedbacks, we cannot comment on how our values compare to natural variability. We argue that our results are substantial in light of the direct forcing of these specific species driven by anthropogenic emissions, or climate feedbacks (stated explicitly in last page of the manuscript), not relative to $CO_2$. We have added a sentence to clarify this.

Modification:
   j. Page 10, lines 29-30: text added: "We note that these estimates are obtained with fixed 2010 meteorology, and therefore we have not assessed the interannual climate variability against which these values can be compared for significance."

---

## Author Comment (AC2) · 15 Nov 2016

**Response to Reviews**

We would like to thank the reviewers for their comments and suggestions. We have addressed these below (in blue below original comment).

**REVIEWER #2**

This paper examined the impacts of historical land use change (LUC) and the associated agricultural emission change (AEC) on ozone and secondary particulate matter between preindustrial and present day. The main conclusion is that LUC+AEC result in increased burden of nitrate but decreased burden of BSOA and ozone. Such changes further induce radiative perturbations which present a strong cooling forcing since 1850. This is a fantastic work and analyses are comprehensive. Some minor revisions are required before the publication.

We thank the reviewer for their positive comments.

1. Some results presented in the study may be model dependent. The authors applied GEOS-Chem (GC) model in their study. Although the GC is a widely used and validated CTM, some inherent characteristics may definitely affect the changes in atmospheric chemistry. For example, to explain why the surface nitrate shows large deviations but tropospheric nitrate burden shows small differences between simulations using 1850 and 2000 anthropogenic emissions, the authors claim that "the increase in surface nitrate from pre-industrial to present-day is controlled more by the rise in anthropogenic NOx emissions than the rise in agricultural ammonia emissions, while the increase in the burden of tropospheric nitrate is driven primarily by the increase in ammonia". Are there any observations supporting such conclusion? Similar problems exist for ozone changes (shown in the detailed comments below). The authors need to discuss the possible uncertainties of these responses and reminder readers that the predicted changes in atmospheric composition is somewhat model-dependent.

The reviewer makes a good point that all modeling results are, to some extent, model dependent (hence the value of multi-model assessments). To our knowledge, none of our results are exceptionally dependent on the use of the GEOS-Chem model, however we allow that such dependencies (on the specific chemistry scheme, on the GMAO meteorology, etc.) may exist. We add a sentence to acknowledge this. Unfortunately we do not have observational constraints over the pre-industrial to present-day to verify our results, and this must therefore be considered purely a modeling study based on our current knowledge of biosphere-atmosphere exchange and atmospheric chemistry.

Modification:

    k. Page 11, lines 19-23: text added: "The simulations analysed in this study were performed with one chemical transport model (GEOS-Chem); the degree to which model-specific treatments of chemical oxidation, aerosol formation, and meteorology may impact the results cannot be assessed here. Thus, additional modelling investigations using alternate model schemes are required to better characterize the uncertainty surrounding the impact of land use change on air quality and climate forcing."

2. The authors performed sensitivity experiments to isolate the impacts of LUE and AEC (Table 2) but did not present those results in their analyses. Based on the qualitative explanations, we can understand that the large enhancement of nitrate is mainly attributed to AEC, the reductions in biogenic secondary

organic aerosols (BSOA) is dominantly driven by LUC, and the decline of ozone burden is a compound result of AEC and LUC, and the impacts of LUC seem to overweigh that of AEC. However, without quantitative numbers, we do not know the individual contributions of LUC and AEC. I suggest that the authors add a new Table to summarize changes in atmospheric composition due to different drivers (LUC, AEC, and LUC+AEC) as indicated in Table 2.

We have expanded Table 3 to include the quantitative differences in the simulations as requested.

Modification:
    l.   Table 3 now separately specifies emissions changes due to LUC, and LUC+AEC

3. Definition of LUC is confusing. Sometimes, LUC refers to LUC+AEC: "The global annual mean tropospheric burden of aerosol nitrate increases almost 4-fold due to historical LUC (Table 4)". In the following sentence, however, LUC refers to land use change alone: "This increase is almost entirely the result of ammonia emissions increases; land use change alone (simulations 1 vs 2; see Tables 1 and 2) increases the tropospheric burden of nitrate by only 1.1%". In addition, the phrase "land use change" is used frequently after the definition of abbreviation "LUC" in the paper. Similar problem exists for 'DRF' and 'BSOA'. Some clean-up work is required for the clarity.

We appreciate the reviewer's suggestion. We defined LUC to incorporate the net results of land use change and the associated agricultural emissions changes (and present only those results in the paper). We have clarified this in the text.

Modifications:
    m.   Page 5, lines 15-16: added text: "We focus our results on the net impacts of land use change along with the associated changes in agricultural emissions (which we collectively refer to as LUC), unless otherwise specified."
    n.   We have replaced most usages of "direct radiative forcing" with DRF and most usages of "land use change" with LUC in the text.

SPECIFIC
1. The title of the paper may be more appropriate as "The Impact of Historical Land Use Change From 1850 to 2000 on Ozone and Secondary Particulate Matter"

We agree, and have made this change.

2. Page 9 Line 2: "where soil NOx emissions increase due to land use change", here NOx emissions are due to AEC instead of LUC. Similar statement in the paper needs to be clarified.

In fact the changes to soil NOx are due to both LUC and agricultural emissions (as shown in Table 3). As we have defined LUC to include both land use change and the associated agricultural emissions, this sentence remains unchanged.

3. Page 9 Lines 2-5: "Ozone production is widely NOx limited under 1850 anthropogenic emissions, and thus the ozone production efficiency of additional soil NOx emissions is considerably higher, and outweighs the impact of elevated deposition velocities for ozone due to LUC" This cannot explain why the burden of ozone is still decreased due to LUC with 1850 anthropogenic emissions.

We believe the reviewer may have misinterpreted the sentence. The purpose of this sentence is to explain the contrast in surface concentrations when using 1850 anthropogenic emissions (vs. 2000 anthropogenic emissions) NOT the difference in the burden. The changes in surface concentrations are modest and localized and translate in both cases to a very small decrease in burden. This is consistent with Figures 7, 8, 9, and 10.

4. Page 9 Line 19: "DRE" means "direct radiative effect" or just typo for "DRF"?

Thank you for catching this. We have added text to define DRE as "direct radiative effect"

5. Figure 5 caption: Changes of soil NOx and ammonia are caused by AEC instead of LUC.

In fact the changes to soil NOx are due to both LUC and agricultural emissions. We have clarified the caption.